# Implementation of a Canine Ergonomic Abdominal Simulator for Training Basic Laparoscopic Skills in Veterinarians

**DOI:** 10.3390/ani13071140

**Published:** 2023-03-23

**Authors:** Luis C. Hincapié-Gutiérrez, Carlos A. Oviedo-Peñata, Manuel A. Rojas-Galvis, Carlos H. Riaño-Benavides, Juan G. Maldonado-Estrada

**Affiliations:** 1OHVRI-Research Group, Faculty of Agrarian Sciences, University of Antioquia, Medellín 050034, Colombia; 2Tropical Animal Production Research Group, Faculty of Veterinary Medicine and Zootechny, University of Cordoba, Monteria 230002, Colombia; 3Latin American Center for Research and Training in Minimally Invasive Surgery Foundation, Bogotá 251008, Colombia

**Keywords:** concurrent validity, construct validity, content validity, laparoscopic surgery, MIS, simulation, surgery skills, veterinary surgery

## Abstract

**Simple Summary:**

The creation of simulation models that enable the training of basic and advanced skills has been a challenge in veterinary medicine. The idea of this study was to test a laparoscopic simulator initially created for the acquisition of an advanced laparoscopic skill, such as an intracorporeal suture for total laparoscopic gastropexy in dogs. The objective was to evaluate and validate the canine ergonomic abdominal simulator for the acquisition of basic laparoscopic skills; it was used by beginners without previous experience in laparoscopic surgery. The beginners’ performance on our proposed simulator was compared with a previously validated simulator. The simulator provides the acquisition of basic laparoscopic skills, reflected in the surgical performance scores, and improves the time metrics, number of movements, and angular displacement. In addition, it provides the acquisition of laparoscopic surgical skills in the same way as a previously validated simulator. This will allow those interested in acquiring both basic and advanced skills to practice on the same simulator.

**Abstract:**

The validity of the CALMA Veterinary Lap-trainer simulator (CVLTS) for training basic veterinary laparoscopic skills was assessed and compared to a simple collapsible mobile box trainer. Ten veterinarian surgeons with no experience in laparoscopic surgery and four experts with at least two years of experience in minimally invasive surgery (MIS) were included. The training curriculum included object transfer, non-woven gauze cutting with curved scissors, and interrupted and continuous intracorporeal sutures, which were practiced on the CVLTS. The initial and final assessments were carried out in both the CVLTS and in a collapsible mobile simulator. These were video-recorded and evaluated by external experts using the Objective Structured Assessment of Technical Skills (OSATS) and a specific scale evaluation in a double-blinded schedule. The time, angular displacement, number, and movement smoothness were recorded using a hands movement assessment system (HMAS). Through a survey, the face validity and content were evaluated. The data were analyzed by a Pearson’s proportions comparison or Mann Whitney U test and a bilateral Student’s *t*-test. The experimental group OSATS, specific scores, and HMAS values, with the exception of the smoothness of movements, significantly improved after training, with no statistically significant differences compared to the expert group. No differences were found between the two simulators. The experts’ and experimental participants’ CVLTS mean score was 4.8. Our data support the CVLTS validations for laparoscopic surgery basic skills training.

## 1. Introduction

Teaching Minimally Invasive Surgery (MIS) is limited in veterinary schools in Colombia, most likely due to the need to prioritize the teaching of conventional surgery, the excessive cost of the equipment and instruments needed for MIS, or both [1]. Although most veterinary schools in North America and Europe have plenty of MIS experts, including board-certified surgeons, there are other schools that lack these experts. The poor knowledge of the advantages of MIS and the scarcity of MIS experts in some veterinary schools could also constitute significant obstacles. The conventional training of the curricula in veterinary surgery requires the presence of an expert tutor to guide the development of skills by novices [2], as suggested for MIS, where skills learning is more than intuitive and differs from conventional surgery [3]. At present, surgeons’ accreditation to perform MIS is framed within the Halstednian learning method, where an expert supervises the trainee’s training protocol and makes a subjective evaluation of their skills according to the teacher’s experience. This scheme is subjective and prone to error bias [4], and the degree of the competence of the trainee in many procedures is not verified [5,6]. Training in laparoscopic surgery requires developing and validating simulators and curricula for training by inexperienced veterinarians. Although there are learning assessment scales for laparoscopic surgery, such as the Objective Structured Assessment of Technical Skills (OSATS) and Global Operative Assessment of Laparoscopic Skills (GOALS) scales, the lack of quantitative measurement instruments affects the precision and results of the evaluation of surgical performance, because surgical competition is multimodal. This type of evaluation allows for the specific assessment of manual skills through metrics [7]. Having quantitative assessment scales would provide comprehensive information on student performance, identify critical factors in the training process, and better discriminate between expert and novice performances [8]. Some simulators that have been developed and validated for training veterinarians in laparoscopic surgery include the Mayo Simulated Endoscopy Imaging (MESI) canine abdominal model [9,10], the SIMULVET^®^ canine laparoscopic simulator (CLS) [11], the standing equine laparoscopic ovariectomy (SELO) simulation model [12], and the CALMA Veterinary Lap-trainer simulator (CVLTS) [7]. MIS training involves the acquisition of basic and advanced laparoscopic skills and their application to given surgical protocols, which have to be trained and their skill acquisition demonstrated before performing the procedure on the live patient [1]. Therefore, skills development should be addressed through training programs that allow the certification of the learning achieved by the trainees [1,13,14]. Furthermore, simulation is crucial in teaching, learning, and assessing the skills needed to develop trained surgical competencies [15] in both conventional surgery and MIS. The cumulative sum learning curves (CUSUM) are a set of tools that help in the certification process of the participants as it enables the evaluation of the surgical performance through repetitions of the same procedure, as evidenced by Pope and Knowles (2014), who determined that after performing 80 laparoscopic ovariectomies, a veterinarian has the surgical competence to reduce the rate of complications [16]. The present work complements a research project that evaluated the use of CVLTS in advanced laparoscopic skills training [1]. The specific aim was to evaluate and validate the CVLTS for acquiring basic laparoscopic skills in novices with no previous laparoscopic surgery experience compared to a previously validated simulator, the Train Anywhere Skill Kit (TASKit^®^, Ethicon Endo-Surgery Cincinnati, Johnson and Johnson, Cincinnati, OH, USA). The data were assessed using the conventional objective scales and by a motion-sensing device that provided the quantitative data.

## 2. Materials and Methods

Experimental, analytical study. Veterinarians with knowledge and training in open surgical procedures, but without previous experience in laparoscopic surgery or simulation procedures (Experimental group, n = 10), and veterinarians trained in MIS with training certifications approved by the Jesus Uson Minimally Invasion Center in Caceres, Spain, and the Clinical Simulation Laboratory of the University of Antioquia in Medellin, Colombia, and who have been performing laparoscopy surgery for more than two years (Expert group, n = 4) were included. The sample size was established by convenience due to the limited availability of experts in minimally invasive surgery in Colombia [17]. All of the participants completed a demographic survey including age, sex, laterality or hand dominance preference, and video game experience.

The CVLTS simulator was described and validated for advanced laparoscopic skills [1] (Figure 1A). Basic skills training was performed on the CVLTS settled on a wooden platform with a 25° degree inclination for exercise adjustments (Figure 2A,C,E). To determine the concurrent validity of the CVLTS simulator, it was compared with the Train Anywhere Skill Kit (TASKit^®^) (Ethicon Endo-Surgery Cincinnati, Johnson and Johnson, Cincinnati, OH, USA) trainer. The TASKit^®^ is a portable and collapsible simulator previously validated for training basic laparoscopic skills [18]. It has a flat base, allowing the different exercises to be attached (Figure 2B,D,F).

Two methods to assess surgical skills were used. The first was the Objective Structured Assessment of Technical Skills (OSATS) [19], which used a list of operational competencies with behavior and performance descriptors that are rated on a performance scale of 1 (low) to 5 (high) Likert-type points (Appendix A). This provides the evaluation of the general psycho-technical aspects of laparoscopic surgery, such as dexterity of both hands (1 = uses one hand; 3 = ignores the non-dominant hand; 5 = uses both hands in a complementary way), thus optimizing the structures’ exposure [20]. A 7-item specific rating scale was also used to assess each exercise’s proficiency and difficulty, including three coordination tasks (transfer), two cutting tasks, and two suturing tasks (interrupted and continuous). To score both scales, a video was recorded and sent for evaluation by double-blind MIS external experts, independent of the research group and external to the study, who assigned a score based on whether the veterinarians performed correctly, incorrectly, or did not perform the steps established to develop each task as previously reported [18,19].

The second evaluation method was through a Hands Movement Assessment System (HMAS) called Iglove (Figure 1B). This system uses electronic inertia sensors attached to each hand by adjustable straps, which record the movements during the exercises. The skill metric that the system uses is: (i) time to complete; (ii) number; (iii) smoothness in movements; and (iv) angular displacement [21]. A biomedical engineer supervised all of the skills evaluations recorded with the HMAS.

To assess the skills of the experimental group members (trainees) before and after training, the participants received a theoretical session guided by the principal investigator (LCH). It included the MIS principles, evaluation methods, HMAS wearing during simulation, ergonomics, and the correct handling of the simulators and instruments. To standardize the participants’ basic entry behavior, they were shown videos of the seven training exercises and the correct way to perform them. After the theory session, the participants were scheduled to perform the initial assessment (I.A.). Before the assessment, instrument handling outside of the simulators was permitted for a few minutes to familiarize the participants with the instruments. In addition, questions about the exercises and instruments were addressed by the same principal investigator before each assessment. Once the evaluation began, no further questions were answered. The participants performed the seven exercises described in Appendix A, first on the CVLTS, and then on the Ethicon TASKit simulator.

Once all the trainees performed the I.A. in both simulators, the training plan was conducted only in the CVLT. All of the experimental group members attended a minimum of three weekly sessions, lasting between 1 and 2 h, until achieving the objectives set for each task, corresponding to the basic laparoscopic skills (Appendix A Requirements). In addition, the experimental group subjects were supervised during each training session, and effective feedback was provided. Emphasis was placed on triangulation, precision, movement coordination, depth perception, the management of two-dimensional space, and technical ergonomic postures during each one of the sessions.

Once the training was finished, the experimental group performed a final assessment (F.A.), first on the CVLTS and then on the TASKit simulator, intending to measure the gain in basic laparoscopic surgical skills. To compare the experimental group’s objective evaluations and metrics, the expert group members performed the evaluations in both simulators only once under the conditions indicated for the I.A. of the experimental group (The expert group did not perform a final assessment by virtue of their expertise). Furthermore, the initial and final assessments were recorded in both simulators, although the training was performed only in the CVLTS simulator.

A basic laparoscopic surgical skill level was established at the beginning and end of the training for the experimental group subjects, analyzing the means of the metrics data obtained through the HMAS and scales video qualification by double-blinded external experts. In all cases, the double-blinded external experts were veterinary surgeons with extensive experience in MIS, who did not participate in the experimental or expert groups, as reported elsewhere [1].

The score concordance with the data of the experimental group was established through data analysis by the external experts. The closer an experimental participant’s score is to the expert group, the greater the agreement reached for a given parameter. On the other hand, the performance scores that the experimental group participants obtained for the tasks performed on the CVLTS and the TASKit simulator at the beginning and end of the training were assessed for possible correlation.

A validated anonymous satisfaction survey [11,12] was conducted to determine the perception of all the study participants once they completed the evaluation of the CVLTS simulator. The survey asked for information about their experience with the CVLTS, focusing on the simulator design, usability, and other aspects of the simulator. The responses were recorded on a 5-point Likert-type scale, where 1 and 5 correspond to a negative and positive perception, respectively (Appendix A).

Before starting the experiment, a pilot test was carried out with an independent volunteer from the study who filled out the demographic survey, signed the informed consent, and performed the exercises in both simulators under the same methods of evaluation of the basic surgical skill. To ensure the integrity of the data obtained during the execution of the study, and that all the sources of associated error were considered, the data generated in the pilot test were run several times in the LAPVet-App Software (Software version 2020, Bioinstrumentation and Engineering Research Group, Medellín, Colombia), and the video was sent to the external experts to be evaluated in the double-blind MIS, who became familiar with the formats and evaluation methodology. In addition, the experimental group members only completed two-hour maximum sessions to avoid mental exhaustion, which could interfere with their skills improvement [15]. As mentioned earlier, the videos provided to the double-blind MIS external experts were coded to guarantee a double-blind assessment.

To determine the concurrent validity, the performance of the individuals in the experimental group was compared between the evaluations on the CVLTS simulators and the TASKit simulators (Ethicon Endo-Surgery Cincinnati, OH, USA) [18]. The test results were correlated with those obtained by the known standard criterion (Taskit^®^) to determine if the same training results were achieved with the CVLTS [11,12,21]. In addition, the results of the initial evaluations of the experimental group and experts were compared to assess the ability of the CVLTS to detect differences between groups with different levels of competence. The data from the first assessment was compared between the veterinarians with no expertise in laparoscopy surgery and the expert surgeons to determine the construct validity [12], which was assessed as indicated in the statistical analysis.

Statistical analysis. Descriptive statistics were performed using measures of the central tendency-deviation and frequency tables, depending on the nature of variables. Normal distribution was assessed through distribution graphs and the Shapiro-Wilk test. The ICC was determined to observe both of the evaluators’ reliability and level of agreement. Differences in the participants’ training plan were determined by Pearson’s comparison of proportions or the Mann-Whitney U test (based on the nature of the variable). The pre- and post-training skill scores were compared using a bilateral Student’s t-test. A Spearman’s correlation test compared the total scores on the CVLTS with the scores obtained on the TASKit simulator. All of the statistical analyses were performed with R Studio statistics software version 3.5.1, with statistically significant differences at *p* < 0.05.

## 3. Results

Demographics. Four men and six women, between 29 and 33 years old, with an average of 2.6 years of experience in open surgery, and an average per participant of 147 procedures performed, were included in the experimental group, all of whom performed conventional surgery, and all were right-handed. Their average training time in the CVLTS was 10 h. Four men, aged over 33 years, right-handed, with seven years of experience in laparoscopic surgery and fifty-eight laparoscopic procedures on average per participant, were included in the experts group.

Face and content validity of CVLTS simulator. In the 5-point Likert-type survey developed to determine the face and content validity of the CVLTS in basic laparoscopic skills, the experts and experimental group obtained an average of 4.8 for the seven survey questions, supporting their complete agreement with the ratings provided (Table 1).

The experimental group significantly reduced their (*p* < 0.01) time, the average number of movements (Figure 3), and the average angular displacement (Figure 4A,B) post-training in both simulators compared to the pre-training values. Similarly, the mean OSATS score increased significantly after the training (*p* < 0.01) (Figure 5). However, no statistically significant differences were found for the mean smoothness in movements in the CVLTS (*p* > 0.05) (Figure 4C,D).

Once the results of the experimental group’s final evaluation were obtained, they were compared with the results of the expert group’s evaluation. The training time, smoothness in movements, number of movements, and angular displacement of the expert group were significantly reduced (*p* > 0.05) compared to the initial evaluation of the experimental group using CVLTS. In the same way, the OSATS scores were statistically higher for the group of experts concerning the initial evaluation of the experimental group (Table 2).

On the other hand, there were no significant differences (*p* > 0.05) in the training time, number of movements, and angular displacement between the final evaluation of the experimental group and the result of the single evaluation carried out on the expert group. The movement smoothness and OSATS score were significantly better for the experimental group after training (*p* < 0.01) compared to the result of a single evaluation of the expert group (Table 3).

When comparing the CVLTS simulators and the TASKit simulator, it was found that the number of movements, the smoothness of the movements, the angular displacement, and the OSATS scores did not significantly differ between both simulators compared to the initial and final assessment (*p* > 0.05). The raining times between the CVLTS and TASKit simulators were statistically different between the initial and final tests (*p* = 0.0014 and *p* = 0.023, respectively) (Table 4).

## 4. Discussion

This study evaluated the face, content, construct, and concurrent validity of the CVLTS and its reliability in acquiring basic laparoscopic skills, as evidenced by the OSATS and HMAS data. The CVLTS was valid for the learning and improving basic laparoscopic surgical skills of veterinary surgeons with no laparoscopy surgery experience. The face, content, construct, and concurrent validation is supported by the high degree of satisfaction given to the CVLT by the experts and experimental group (Table 1), the results of the initial and final assessment (Table 2), the comparisons between the experimental and expert groups (Table 3), and the comparisons between both simulators (Table 4). The basic laparoscopic skills of the experimental group significantly improved after the completion of the training plan, with no statistically significant difference compared to the experts at the final assessment (Table 2 and Table 3). In addition, the time required to complete the training plan was significantly lower using the CVTLS compared to the TASKit simulator (Table 4). We used the traditional validation methodology developed by the American Psychological Association (APA) [7,11,12] because all the works consulted in the scientific literature on the validation of simulators in the field of veterinary medicine have used this type of validation. However, we are aware that there are other modern validation frameworks.

The CVTLS was developed to offer a valuable and affordable educational tool to train veterinarians in laparoscopy surgery. The initial learning curve phase for a surgeon who wishes to learn laparoscopic surgery should begin with simulated models outside the operating room, enabling deliberate and repeated exercises in safe and controlled environments [11,19,22]. Simulation increases novices’ skills in MIS, which should be transferable to their performance in the operating room [23,24,25,26]. The CVLTS’s face and content validity are supported by the expert scores given to the experimental group (mean rating of 4.8 for all survey questions (Table 1), indicating that they were satisfied with the ratings provided [27]. This finding agrees with the report by Elarbi et al. (2018), who rated 4.4 points on the Likert scale using a laparoscopic ovariectomy in a standing horse model [12].

The CVLTS’s construct validity is supported by its ability to differentiate between expert surgeons’ scores and veterinarians with no expertise in laparoscopy surgery at the first assessment (Table 2), similar to the report by Elarbi et al. [12]. Construct validity is considered the most critical step in simulator models’ validation due to the importance of supporting that it works for the purpose it was created [7,28]. Accordingly, an increase in the OSATS scores and reduced movement metrics after training completion were found in the experimental group when performing on the CVLTS (Figure 3, Figure 4 and Figure 5). In addition, statistically significant differences were found in the experimental group compared to the experts in the pre-training evaluation, but not in the post-training evaluation, with the exception of the OSATS scores and mean smoothness in movements at the final assessment (Table 3). These results differ from the report by Fransson et al. [9]. We suggest that this difference could be explained because the expert surgeons did not train on the CVLTS, which may have influenced their familiarity with the simulator when performing exercises. In contrast, the experimental group participants remained motivated by the opportunity to gain experience in basic skills in veterinary laparoscopy surgery. This bias could be eliminated by subjecting the members of the experimental group to training in the same way as the participants of the experimental group. Unfortunately, the availability of time on the part of the experts limited the exposure during the number of hours required for the training. On the other hand, the internal motivation of the experts regarding training in basic skills is minimal, especially when laparoscopic procedures are already performed in their daily clinic.

The HMAS focuses primarily on assessing the participant’s ability to perform tasks as quickly as possible with movement economy. Their limiting factor is that the attention level and how the tasks are performed are not measured directly, which are aspects currently assessed using OSATS. Therefore, quickly doing training tasks does not always guarantee that they are completed well. Thus, it is critical to consider involving as many different evaluation methods as possible when looking for reliable results when investigating the acquisition of technical skills.

The experimental group participants’ skills achievements agree with previous reports using the TASKit simulator [29]. The CVLTS’s ergonomic design probably explains the lack of statistically significant differences in the average smoothness in movements after training, in agreement with the report by Fransson and Ragle (2010), where training group scores significantly improved between their first and second evaluation and the participants performed significantly better than the scores of the expert group in some tests [10].

No statistically significant difference was found for the smoothness in movements, angular displacement, number of movements, and OSATS values when comparing the initial and final assessment results obtained by the experimental group in both simulators, with the exception of the training time (Table 4). These results demonstrate a concurrent validity between the CVLTS and TASKit simulator [29,30]. Accordingly, the CVLTS could be used in training plans to acquire basic laparoscopic skills in veterinary MIS surgery. Furthermore, the training time significantly improved using the CVLTS, probably due to the greater comfort from the visual aspect of the screen (e.g., sharpness, color, size) when performing exercises. In addition, the training curriculum allowed the participants to perform evaluations on both simulators, first on the CVLTS and then on the TASKit simulator. This fact could imply that muscular and visual fatigue occurred when performing exercises in the latter simulator. In future studies, this factor could be included by randomizing the turn of simulator use.

Limitations of the study. First, the number of experts and veterinarians in the experimental groups was small due to the limited availability of MIS experts in the Metropolitan area of Medellin. Second, the lack of randomization for using the simulators.

In conclusion, using a simulator to train and acquire basic MIS skills allows veterinary surgeons with no previous MIS experience to acquire and significantly improve their laparoscopic skills. Simulators such as the CVLTS represent one of the first models involving qualitative and quantitative evaluations, representing a critical advantage for training OSATS in veterinary laparoscopic skills training centers. The HMAS metrics and OSATS scores proved helpful for basic training on the CVLTS. The simulator allowed the differentiation of the experience level between the experimental group at the beginning of the training curriculum with the expert group. The improvement of the experimental group in all variables resulted in the acquisition of basic MIS skills at the end of the training, which was not statistically different from those of the expert group.

## 5. Conclusions

Our work demonstrates the face, content, construct, and concurrent validity of the CVLTS for training veterinarians in basic skills acquisition for MIS, comprising qualitative and quantitative evaluations. Similarly, this simulator can be used under self-guided training or the supervision of an expert as it involves using the HMAS, which provides quantitative data of the training time, number of movements, angular displacement, and roughness of movements immediately after training. This system allows the immediate qualification of an expert in combination with the OSATS-type evaluation scale, resulting in a better training program.

## Figures and Tables

**Figure 1 animals-13-01140-f001:**
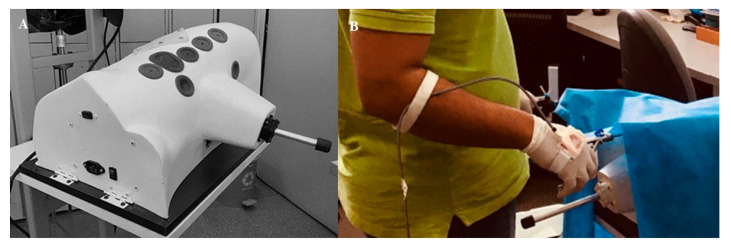
(**A**) CALMA Veterinary Lap-trainer simulator. (**B**) Hands movement assessment system (HMAS).

**Figure 2 animals-13-01140-f002:**
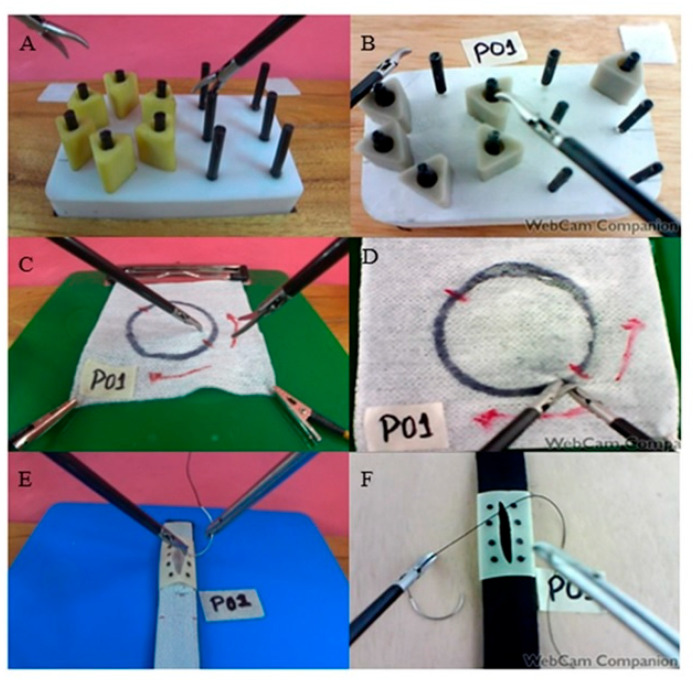
(**A**,**C**,**E**): Interior image of CVLTS, and. (**B**,**D**,**F**): TASKit^®^ simulator.

**Figure 3 animals-13-01140-f003:**
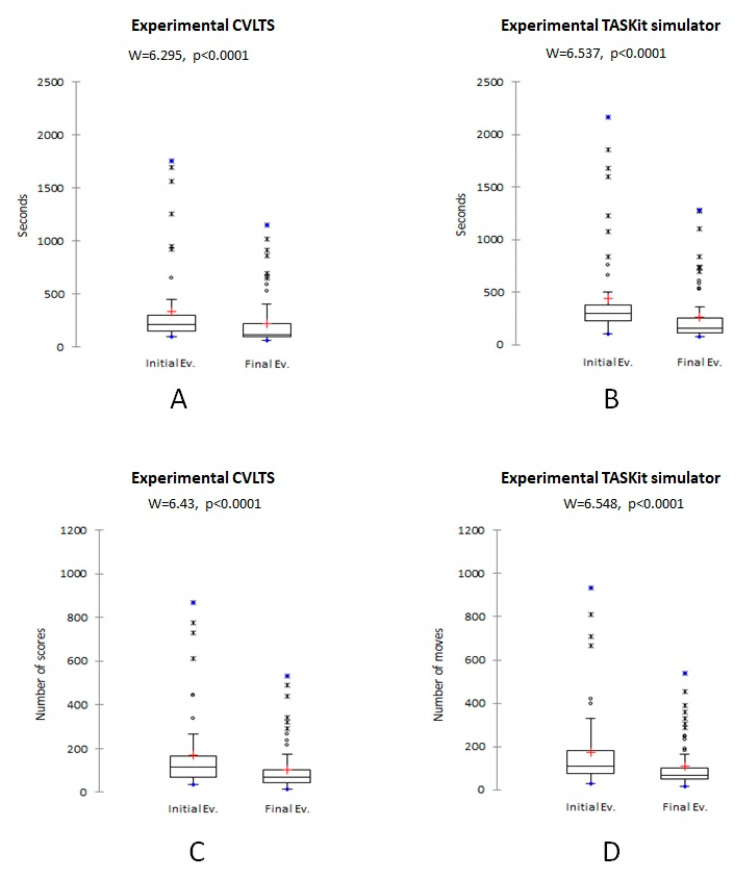
Comparison of the experimental group’s mean training time (**A**,**B**) and the mean number of movements (**C**,**D**). Data are presented for CVLTS (**A**,**C**) and TASKit (**B**,**D**) simulators.

**Figure 4 animals-13-01140-f004:**
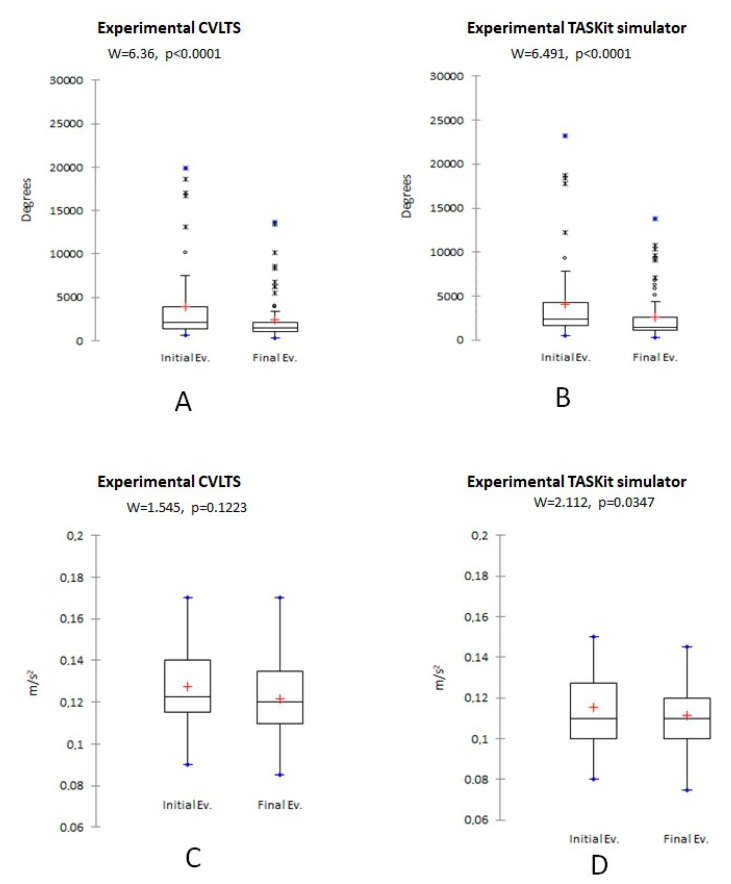
Comparison of the experimental group mean angular displacement (**A**,**B**) and mean smoothness in movements (**C**,**D**). Data are presented for CVLTS (**A**,**C**) and TASKit (**B**,**D**) simulators.

**Figure 5 animals-13-01140-f005:**
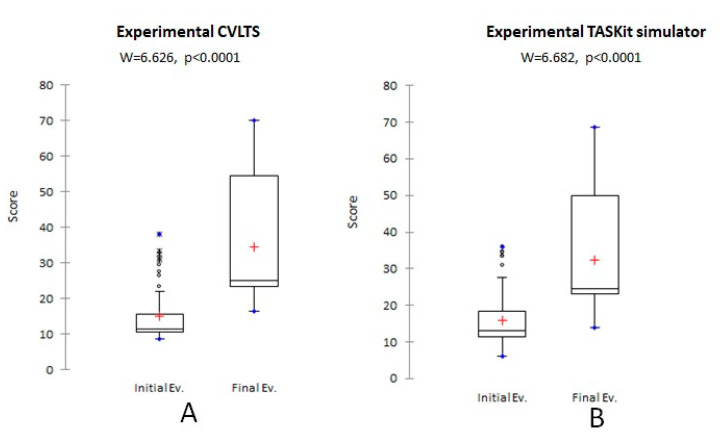
Comparison of the experimental group average OSATS scores using (**A**) CVLTS and (**B**) TASKit simulators.

**Table 1 animals-13-01140-t001:** Face and content validity of the CVLTS.

Questions about CVLTS	Score *					Mean	Min–Max
	1	2	3	4	5		
It is realistic, didactic, and appropriately sized for training in basic laparoscopic skills.				2 (14.3%)	12 (85.7%)	4.86	4–5
It has a clear, pleasant, and colorful image quality.				1 (7.1%)	13 (92.9%)	4.93	4–5
Performance on exercise range			1 (7.1%)	3 (21.4%)	10 (71.4%)	4.64	3–5
It is helpful for veterinary training students.				2 (14.3%)	12 (85.7%)	4.86	4–5
It is helpful for training in veterinary laparoscopic surgery				1 (7.1%)	13 (92.9%)	4.93	4–5
Help to improve my laparoscopic skills and apply them to my patients				2 (14.3%)	12 (85.7%)	4.86	4–5
Inclusion in laparoscopy training programs for veterinary students before practice in the operating room?				2 (14.3%)	12 (85.7%)	4.86	4–5

* Rating: 1 = Very negative; 5 = Very positive.

**Table 2 animals-13-01140-t002:** Construct validity. Comparisons between the initial evaluation of the experimental group and expert group (Q1–Q3) Mann-Whitney U test (α = 0.05).

Variable	Experimental CVLTS	Expert Group	*p*-Value
Time of training plan (seconds)	214.5 (154.7–301.2)	132 (99.7–229.7)	0.0014
Mean smoothness in movements (m/s^2^)	0.1225 (0.115–0.14)	0.14 (0.12–0.151)	0.0261
Mean movements (number)	113.5 (69.2–167.4)	68.25 (42.9–107.5)	0.0059
Mean angular displacement (degrees)	2104.7 (1366–3956)	1517.6 (1016–3170)	0.0368
OSATS (Score) *	11.5 (10.5–15.5)	20.75 (18.5–50.37)	<0.0001

* ICC = 0.99, *p* < 0.01, meaning statistically significant agreement between external experts for OSATS assessment.

**Table 3 animals-13-01140-t003:** Construct validity. Comparisons between the final evaluation of the experimental group and expert group (Q1–Q3) Mann-Whitney U test (α = 0.05).

Task	Experimental CVLTS	Expert Group	*p*-Value
Time of training plan (seconds)	120 (100–223.7)	132 (99.7–229.7)	0.6369
Mean smoothness in movements (m/s^2^)	0.12 (0.11–0.13)	0.14 (0.12–0.151)	0.0004
Mean movements (number)	68 (45.5–103.1)	68.25 (42.9–107.5)	0.9874
Mean angular displacement (degrees)	1507.7 (1016–2120)	1517.6 (1016–3170)	0.7861
OSATS (Score)	25 (23.5–54.37)	20.75 (18.5–50.37)	0.0002

**Table 4 animals-13-01140-t004:** Concurrent validity. Comparisons of the results obtained from the experimental group between CVLTS and TASKit simulators (Q1–Q3) Mann-Whitney U (α = 0.05).

Variable	Experimental CVLTS Simulator	Experimental TASKit Simulator	*p*-Value
Time of training plan (seconds)			
Initial	214.5 (154.8–301.3)	300 (231–381)	0.0014
Final	120 (100–223.8)	154.5 (117.3–252.5)	0.023
Mean smoothness in movements (m/s^2^)			
Initial	0.12 (0.11–0.14)	0.11 (0.1–0.13)	0.7532
Final	0.12 (0.11–0.14)	0.11 (0.1–0.12)	0.905
Mean movements (number)			
Initial	113.5 (69.3–167.4)	111 (77.8–181.8)	0.7894
Final	68 (45.5–103.1)	67.75 (52–102.3)	0.6406
Mean angular displacement (degrees)			
Initial	2105 (1366.1–3955.8)	2341 (1707.5–4284.2)	0.5814
Final	1508 (1016.2–2120.2)	1493 (1168.6–2568.4)	0.6708
OSATS (Score)			
Initial	11.5 (10.5–15.5)	13 (11.5–18.3)	0.158
Final	25 (23.5–54.4)	24.5 (23.1–49.9)	0.1005

## Data Availability

Most of the data is presented in the article. Excel databases, statistical analysis files, and graphic files -photographs and videos- are available to the scientific community upon request.

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
