# Peer review of "Implementation of a Canine Ergonomic Abdominal Simulator for Training Basic Laparoscopic Skills in Veterinarians"

_animals, 2023, doi:10.3390/ani13071140_

Round 1

Reviewer 1 Report

Interesting manuscript. Validations like this increase the teaching of veterinary medicine and increase the learning capacity of our students.

Title: improve - Evaluating and validating? for example - System implementation...

abstract: ok

Introduction: improve the text. The authors do not put paragraphs. More information on learning curves in this type of material is lacking.

Material and Methods: ok

Results: ok

Discussion: ok

Conclusion: ok

Others: ok

Author Response

Add comments in a PDF

Reviewer 2 Report

the paper is excellent and provide a good opportunity to improve mininvasive surgery. well done

Author Response

Add comments in a PDF

Reviewer 3 Report

In general your study is interesting but you need to clarify your materials and methods. Please see specific comments within the document, but in general your inclusion criteria is unclear and the participation of experts vs novice is confusing. You also have clear contradictions in your text, please correct.

Please review the English as there are places were it is very difficult to understand what you are trying to say. Also please follow journal guidelines particularly when it come to subheadings. The way you have your headings now makes reading very difficult and makes the lack of English more obvious.

Line 20: Other previously validated what?

Line 26: added to what? doesn´t make sense

Line 28: Please specify

Line 46: prioritize the teaching of

Line 47: appropriation means acquisition, I don´t think this is what you mean to say

Line 48: add some here as there are veterinary schools that have plenty of MIS experts, including board certified surgeons. In alternative, citation of credible source for this broad of a statement

Line 50 – 52: Very confusing, does not make sense please rewrite

Line 52: "surgeons' skills accreditation to perform MIS" does not make sense, do you mean a surgeons accreditation to perform, or surgeons MIS skills are accredited?

Line 62: citation required for this statement

Line 76: first time mentioned, should be written in full

Line 78: which previously validated simulator?

Line 90-91: which characteristics? what where the criteria of inclusion with regards to these characteristics?

Line 98 – 102: Does make sense. I am assuming this is the validated simulator that was used as control but this is very confusing and not in any way clear. Please rewrite.

Line 103: I am assuming this is a subheading. Please make the subheading clear in accordance with journal guidelines.

Line 119: the same expert trainers that participated in the evaluation of the models? Did they self assess? How was this controlled?

Line 129: Again which expert evaluators? Where they the same as the trainers? The same as the participants?

Line 145: are they the same experts referred to above? If so this should be stated beforehand

Line 147: earlier it was stated that only beginners did this part of the evaluation. Did all participants do the IA and the training? This must me made clear.

Line 150: Confusing, was this the basic training or was this completed after the basic training? If so, what was the basic training?

Line 161: how was the score compared? average? one trainee to one expert? this must be specified.

Line 166: from reading this, it seems that the IA and FA was only applied to novice surgeons. However, initially it was implied that ALL participates completed the IA and FA. This needs to be clarified. Who did what when?

Line 173-176: This does not indicate bias control, it does show inicial assessment and correction as needed. It also shows that evaluating experts were first familiarized with what they were accessing. This does not control for bias

Line 185: Please rewrite, not clear, are you defining concurrent validity?

Line 187: Yes, this is a definition, but how was this done?

Line 205: 58 in total or 58 each expert?

Line 238: After or before the training period? who participated in the training period. This is very confusing and unclear.

Line 249: In line 178 it is stated that training was only performed in CVLTS simulator, when and how was training completed in the TASKit simulator? Again both beginners and experts where included?

Line 260 – 264: From here it seems that only the novice veterinarians where trained with the simulators and then the FA of novice vets was compared with the "FA" of experts. Is this the correct method that was followed? If that is the case then the materials and methods section does not reflect this. Please take a close look and correct.

Line 265 – 266: again in line 178 it is stated that only the CVLTS was used in training.

Line 278: in line 209 you state that the "experts and the experimental group obtained an average of 4.8 for the seven survey" however here you state that experts gave to the experimental group? This does not make sense.

Line 293: this is the first time a statement regarding expert inclusion in the training program is made. It should be much clearer in the text.

Line 325: how does this create bias? How would you combat this in the future?

Author Response

Add comments in a PDF

Round 2

Reviewer 3 Report

I want to thank the authors for the time taken to improve the manuscript. I have a few, minor, corrections that mainly have to do with English.

Ln 23: validity vs validities

Ln 24: was vs were

Ln 24: to a simply? do you mean simple or simpler?

LN 34 -35: "Through a survey, it was evaluated face validity and content" the use of"it was" is incorrect

Ln 125: "It provides evaluating general" this does not make sense. Do you mean it allows for evaluation? Or General evaluation? or something else?

Ln 156: trainers or trainees?

Ln 291: This is data for the experimental group, yes? this should be made clear in the table heading

Author Response

Attached letter of responses to the reviewers.
